# The Synergistic Effect of Thiamethoxam and Synapsin dsRNA Targets Neurotransmission to Induce Mortality in *Aphis gossypii*

**DOI:** 10.3390/ijms23169388

**Published:** 2022-08-20

**Authors:** Xueting Qu, Sijia Wang, Guangze Lin, Mingshan Li, Jie Shen, Dan Wang

**Affiliations:** Department of Plant Biosecurity and MARA Key Laboratory of Surveillance and Management for Plant Quarantine Pests, College of Plant Protection, China Agricultural University, Beijing 100193, China

**Keywords:** RNA interference, neonicotinoid, *synapsin*, pest control, nanocarrier

## Abstract

Sublethal doses of insecticides have many impacts on pest control and agroecosystems. Insects that survive a sublethal dose of insecticide could adapt their physiological and behavioral functions and resist this environmental stress, which contributes to the challenge of pest management. In this study, the sublethal effects of thiamethoxam on gene expression were measured through RNA sequencing in the melon aphid *Aphis gossypii*. Genes regulating energy production were downregulated, while genes related to neural function were upregulated. To further address the function of genes related to neurotransmission, RNA interference (RNAi) was implemented by transdermal delivery of dsRNA targeting *synapsin* (*syn*), a gene regulating presynaptic vesicle clustering. The gene expression of *synapsin* was knocked down and the mortality of aphids was increased significantly over the duration of the assay. Co-delivery of syn-dsRNA and thiamethoxam reversed the upregulation of *synapsin* caused by low-dose thiamethoxam and resulted in lethality to melon aphids, suggesting that the decreased presynaptic function may contribute to this synergistic lethal effect. In addition, the nanocarrier star polycation, which could bind both dsRNA and thiamethoxam, greatly improved the efficacy of lethality. These results increase our knowledge of the gene regulation induced by sublethal exposure to neonicotinoids and indicated that *synapsin* could be a potential RNAi target for resistance management of the melon aphid.

## 1. Introduction

Neonicotinoids are effective neuroactive insecticides. All neonicotinoids share the same mode of action: acting as an agonist of the nicotinic acetylcholine receptor (nAChR) [1,2]. However, neonicotinoids not only cause lethal effects in insects, but also sublethal effects, due to their widespread use in the field [3]. With exposure to sublethal doses of insecticide, insects can experience a variety of physiological, developmental, and behavioral disruptions, which results in profound impacts on pest control and the agroecosystem.

Insects have hormesis to low-dose insecticides, which means they have a biphasic dose response [4]. On the one hand, low-dose exposure to insecticides can stimulate biological processes that increase survival and reproduction. For instance, sublethal doses of imidacloprid stimulate reproduction in aphids [5,6,7]. On the other hand, sublethal doses of insecticides can cause reduced reproduction, decreased longevity, delayed development, and compromised locomotion or learning in pest insects, beneficial insects, as well as non-target organisms [8,9,10]. For examples, sublethal doses of neonicotinoids cause reduced reproduction, body weight, honeydew excretion, and longevity in aphids [11,12]. In beneficial insects, such as honeybees and bumblebees, defects in foraging [13,14], circadian rhythms and sleep [15,16,17], learning and memory [18,19], and abnormal neural development [19] have been detected. Similarly, chronic low-level exposure to imidacloprid perturbs mitochondrial function in the brain, lipid droplet accumulation, vision, and movement in *Drosophila melanogaster* [20]. These effects could participate in the emergence of insecticide resistance and cross-resistance in pest insects. One resistance mechanism is metabolic detoxification by upregulation of detoxifying enzymes, including cytochrome P450 monooxygenases, glutathione-S-transferases, and esterases [21,22,23,24,25]. Point mutation of nAChRs is another factor associated with resistance to neonicotinoid insecticides [26,27]. Genes related to mitochondrial function, lipid metabolism, neuronal function, and immunity are the underlying molecular mechanisms of these physiological and behavioral disruptions [20,28,29,30,31].

Studies have shown that knockdown of detoxification genes, olfactory associated genes, cuticular protein family genes, and so on with RNA interference (RNAi) increases susceptibility to neonicotinoids [32,33,34], showing potential for resolving insecticide resistance. RNAi is a sequence-specific post-transcriptional gene silencing phenomenon [35,36], by which double-stranded RNA (dsRNA) or small interfering RNA (siRNA) silences specific genes, and developmental disorders and/or mortality are produced. RNAi has been shown great potential for the development of novel pest management strategies [37,38]. Whether co-delivery of neonicotinoids and dsRNA targeting genes in neurotransmission can achieve lethality in pests at low doses of insecticides is unclear.

The melon aphid (also called cotton aphid) *Aphis gossypii* (Glover) is a worldwide polyphagous pest. It has developed resistance to thiamethoxam (TMX), a second generation of neonicotinoids, and cross-resistance to other neonicotinoids [39,40,41]. In this study, we found that the melon aphid had a strong response in its gene expression to a sublethal dose of TMX. Genes related to neurotransmission were largely upregulated. Specifically, the dsRNA targeting *A. gossypii synapsin* (*syn*) knocked down the expression of *synapsin*, a gene regulating synaptic vesicle clustering, and increased the susceptibility to low-dose TMX. This study provides a potential gene target to affect melon aphid populations, using a neonicotinoid-RNAi-based pest control strategy.

## 2. Results

### 2.1. Genes Related to Metabolism and Neural Function Were Differentially Expressed

To better understand the molecular changes after exposure to sublethal doses of TMX, RNA sequencing was performed and transcripts were annotated in the GO terms database. Comparing the aphids exposed to the LC50 dose of TMX and the untreated control, 1106 genes were upregulated, while 699 genes were downregulated. The biological functions of these differentially expressed genes (DEGs) were mainly involved in the metabolism process and cellular process. In particular, synapse-related genes were categorized in the cellular function category, indicating an important role of TMX in regulation of neural function (Figure 1A). Then, genes related to energy production and neural function were analyzed. Five genes in energy production were upregulated, while 30 genes were downregulated (Figure 1B, Appendix A), indicating changes in energy metabolism. In the neurofunctional genes, 63 genes were increased and 13 were decreased (Figure 1C, Appendix A), suggesting acceleration of neurotransmission.

To validate the transcriptomic data, mRNA levels of 12 neurofunctional genes were selected for a qPCR test (Figure 1D). Eleven upregulated genes showed a similar expression pattern to the DEGs analysis. These genes were involved in various processes of neurotransmission, including neurotransmitter release (*synapsin*, *Eps15-like 1*, *Intersectin-1/Dynamin associated protein 160* (*Dap160*), *dynamin*, *stoned-B*, *Synaptotagmin*, and *Endophilin-A*), adhesive links between pre- and post-synaptic terminals (*Neuroligin-1*), neurotransmitter receptors (*octopamine receptor beta-2R* and *AChR**α-like 1*), and degradation of neurotransmitters (*Acetylcholinesterase*). *V-type*
*proton ATPase*
*subunit*, which is associated with vesicular trafficking and energy supply, is an example of a downregulated gene.

### 2.2. Expression Profiles of Synapsin at Different Stages and Tissues

Numerous studies have shown that synapsin is a key actor in synaptic function and plasticity [42]. Many neurological disorders are determined by the expression levels of synapsin [43]. It is also a sensitive marker to trace brain structure and verify synaptic disorders in insects [44,45,46,47]. Therefore, we took the sole *synapsin* of melon aphid as an example, to study the neuronal disorders caused by TMX. We first compared the protein sequences in insects. The neighbor-joining (NJ) phylogenetic tree was generated using full-length protein sequences of synapsin or predicted synapsin protein sequences from the transcriptome database of diverse species (Appendix A). The insect synapsins from the same order were clustered into one group, while the protein from the non-insect outgroup species *Tetranychus urticae* was in a separate group, implying that synapsin proteins were relatively conserved in insects. *A. gossypii* synapsin was clustered in the Hemiptera group, closest to the *Melanaphis sacchari* synapsin.

To address the function of *synapsin* in melon aphids, the expression pattern was examined. Compared with the first instar nymphs, *synapsin* was highly expressed in the adult stage (Figure 2A). As synapsin is a synaptic vesicle-associated protein, we further examined if the head was a highly expressed tissue. Compared with the adult trunks, the expression of *synapsin* in adult heads was much higher (Figure 2B). In the pea aphid, *synapsin* shows a high expression in the retina and protocerebrum [47]. Thus, we speculated that the aphid *synapsin* played an important function in the nervous system.

### 2.3. Knockdown of Synapsin Led to Death of the Melon Aphid

The predicted *A. gossypii* synapsin protein had two conserved domains. The N-terminal synapsin domain suggests its conservative function in the regulation of neurotransmitter release at synapses, and the CPSase_L_D2 superfamily domain suggests its carbamoyl-phosphate synthase activity that catalyzes ATP-dependent synthesis of carbamoyl phosphate (Figure 3A). Thus, two independent dsRNAs were designed (Figure 3A), which targeted the C-terminal (syn-dsRNA-1) and the N-terminal (syn-dsRNA-2), respectively, to test the function of *synapsin* in the melon aphid. syn-dsRNA was dropped onto the dorsal abdomen of third instar nymphs (Figure 3B).

As the nanocarrier SPc bound syn-dsRNA to form a complex and could stabilize syn-dsRNA in vivo (Appendix A), we compared the relative gene expression with and without SPc application. A droplet containing 250 ng syn-dsRNA was applied. As a control, the same amount of EGFP-dsRNA had no effect on the mRNA level of *synapsin*. Without SPc, *synapsin* was detected as decreasing only at 12 h after syn-dsRNA delivery (Figure 3C). In the presence of SPc, the relative expression level of *synapsin* was 23.6% compared with EGFP-dsRNA samples at 12 h (Figure 3C) and 41.8% at 24 h (Figure 3D) in the syn-dsRNA-1 delivered aphids. Compared with the non-SPc treated aphids, the RNAi efficacy was increased by 66% and 48%, respectively. Both syn-dsRNA-1 and -2 resulted in the interference of endogenous *synapsin* gene and the RNAi efficacy was improved by SPc.

With the effective silencing of *synapsin*, mortality was recorded. The addition of the surfactant APG showed no overt increase in the mortality of melon aphids and was comparable to other control groups without the addition of APG (Figure 4B). Control EGFP-dsRNA showed a minimal, non-significant increase in the mortality (27.5%) of aphids when forming a complex with the nanocarrier SPc (Figure 4B,D). When treated with 250 ng syn-dsRNA-1 without SPc, the mortality was 38.9% after four days, and the mortality reached 66.3% with SPc (Figure 4B). When the amount of syn-dsRNA was 500 ng, the mortality was increased to 72.5% after four days, without SPc. When SPc was applied, the mortality was greatly increased, to 69.7% within two days. The highest lethality rate reached 93.3% within four days (Figure 4D). syn-dsRNA-2 treated aphids showed similar results (Figure 4C,E). At each time point, the application of syn-dsRNA/SPc complex showed higher mortalities than syn-dsRNA (Figure 4B–E), indicating an improvement of lethality with SPc. These data indicated that the amount of syn-dsRNA was positively correlated with the mortality of aphids, and the nanocarrier increased the mortality caused by syn-dsRNA.

To better understand the mechanism that led to death of the melon aphid with syn-dsRNA, we examined the behavior of survivors, since *synapsin* regulates neurotransmission. The melon aphid was not as active as other insects. They were largely immobile, with an occasional stretch of the legs. When putting the melon aphids upside down, they rolled back over quickly (Appendix A). In contrast, when the syn-dsRNA-treated aphids were upside down, they went into paralysis and could not roll over (Appendix A). Aphids with such phenotypes eventually died. This behavior suggested that the neuronal response was most likely a result of knockdown of *synapsin*, which eventually lead to the death of the insect.

Furthermore, the fecundity was recorded for the survivors. The number of offspring laid by each adult showed no difference among EGFP-dsRNA-, syn-dsRNA-1-, and syn-dsRNA-2-treated melon aphids and was comparable to the untreated controls (Appendix A). Therefore, syn-dsRNA did not affect the reproductive capacity of melon aphids.

### 2.4. Co-Delivery of syn-dsRNA and Thiamethoxam Caused Efficient Lethality to Melon Aphids

Since the expression of *synapsin* with low-dose TMX was increased, we wondered if knockdown of *synapsin* at low-doses of TMX could induce a higher lethality. Allowing for delivery of TMX, we used the spray method in this assay (Figure 5A). At the concentration of 5.249 mg/L TMX, 50% aphids were dead at 48 h (red dot in Figure 5B). A combination of TMX and SPc increased the lethality to 88.3% (orange dot Figure 5B). This enhanced toxicity of TMX with SPc is consistent with a previous study [48]. Using the spray method, 43.3% aphids were dead at 48 h with a syn-dsRNA/SPc complex (blue dot Figure 5B). Joint delivery of syn-dsRNA and TMX showed significantly increased lethality at 24 h and all aphids were dead at 48 h (green dots Figure 5B). To verify these results, a lower dose of TMX (1 mg/L) was applied. Indeed, SPc increased the lethality of TMX at all time points. The formulation of TMX/SPC/syn-dsRNA resulted in over 90% lethality at 72 h (Figure 5C). These results indicated that a low dose of TMX could achieve high lethality of melon aphids in combination with syn-dsRNA and nanocarriers.

Next, the expression level of *synapsin* was tested. After exposure to low-dose TMX (1 mg/L), the expression of *synapsin* was upregulated at 48 h (Figure 5D), similarly to that with the higher concentration (5.249 mg/L in Figure 1D). syn-dsRNA more strikingly decreased the *synapsin* level than exposure to TMX (Figure 5D). This result suggested that this synergistic effect on lethality was probably due to the decrease of *synapsin,* by which the adaptation of neuroexcitability to low-dose TMX was disturbed.

### 2.5. The Upregulation of Synapsin as an Acute Response to Thiamethoxam

To further analyze the correlation of *synapsin* with TMX, we measured the *synapsin* levels over a period of time after exposure to TMX. One day after the exposure, the expression level was comparable to the untreated control. Twelve hours later, the expression was upregulated. At sixty hours and beyond, however, the expression was reduced (Figure 6). We speculated that the upregulation of *synapsin* was an acute adaptation to TMX.

## 3. Discussion

In this study, we report for the first time that when knocking down *synapsin* using RNAi, significant mortality can be induced in melon aphids. *synapsin* is enriched in the nervous system, a consistent expression pattern in insects (Figure 2) [47,49,50]. The expression pattern of *synapsin* is able to display the structure of the central nervous system [44,45,46,47]. In both invertebrates and vertebrates, phosphoprotein synapsin, together with the endocytic scaffolding complex, regulates synaptic vesicle clustering, by tethering synaptic vesicles to the actin cytoskeleton, thus facilitates synaptic transmission [42]. Null mutants of the fruit fly *Drosophila melanogaster* show impaired learning and memory, and activity-dependent vesicle recycling and synapse formation [51,52,53]. Knockout mice exhibit a number of behavioral defects, such as altered synaptic transmission, seizures, and depression [54,55], suggesting its crucial role in synaptic plasticity. The paralysis of the melon aphid (Appendix A) suggests a physiological response to the lack of neurotransmitter release by knockdown of *synapsin*. However, *synapsin* is not essential for neuronal development and none of the *synapsin* deletions, such as in the fruit fly and mouse, cause lethal effect [51,54,55].

Several possibilities may explain the lethality to the melon aphid of a lack of *synapsin*. First, during intense stimulation, the excitatory postsynaptic current or potential drops fast and is restored slowly with loss of *synapsin*, indicating that synaptic vesicles are easily depleted and difficult to replenish [54,56]. Continuous stimulation leads to exhaustion of synaptic vesicles and eventually death. Second, synapsin, in complex with Dap160 and Eps15, serve as scaffolding proteins for the clustering of synaptic vesicles [57,58,59,60]. Genetic deletion of both Dap160 and Eps15 proteins in the fruit fly causes severe developmental defects and larvae do not survive through to the pupal stage, similarly to each single null mutant [58], indicating the essential role of this complex for survival. Third, synapsin I is associated with neuronal survival. synapsin I promotes neuronal survival in neonatal mice [61], while synapsin I deletion impairs the survival of newborn neurons [62] and increases neuronal loss in aged mice [63], indicating its important function in cell survival. Thus, the melon aphid could be more vulnerable to reductions of *synapsin* and suffer in their response to environmental stimulation.

In cholinergic synapses in insects, the excitatory neurotransmitter acetylcholine (ACh) is released presynaptically and binds to the postsynaptic receptor nAChR, causing opening of the ion channel for neurotransmission. Neonicotinoids can irreversibly bind to nAChR and block synaptic transmission at the postsynaptic membrane [1,2,64]. In contrast, the presynaptic action potentials are still evoked when the excitatory postsynaptic potentials are blocked by imidacloprid or clothianidin [65,66]. In this study, activation of neurofunctional genes was detected when the melon aphids were exposed to sublethal doses of TMX (Figure 1), suggesting a hyperexcitation of neurons. Intriguingly, the expression of *synapsin* was reduced several hours later, suggesting that the hyperexcitation of neurons may be an acute response. In honey bees, exposure to a sublethal dose of thiamethoxam or imidacloprid also leads to a reduction of *synapsin* expression [19,67], showing the regulation of neonicotinoids in neurophysiology through *synapsin* expression. As knockdown of *synapsin* causes lethality to melon aphids, and as the recombination of syn-dsRNA and TMX accelerates this lethality, we suggest that the blockage of both pre- and post-synaptic functions could be the cause of this synergistic lethality.

Insecticidal activity can be greatly improved by nanocarriers. Nanoparticles have been developed to carry dsRNA and insecticides into the insect body and efficiently result in death, because nanoparticles can not only promote dsRNA, to penetrate into cells across the insect cuticle and protect dsRNA from degradation in vivo [37,38,68], but also act as insecticide nanocarriers, to decrease the particle size of drugs to the nanoscale and increase their efficacy [69,70,71]. In this study, the nanoparticle SPc binds syn-dsRNA and protects it from degradation (Appendix A), consistently with previous reports [72]. The nucleic acid binding ability of SPc is due to electrostatic interaction of its cationic shells with negatively-charged nucleic acids [73]. This nanocarrier-mediated RNAi works in many insects [73,74]. Additionally, nano-sized TMX with SPc, through electrostatic interaction, enhances the delivery efficiency of TMX, as well as the contact and stomach toxicity against green peach aphid [48]. Therefore, an insecticide–nanocarrier–RNAi combination could be a great potential strategy for pest management.

## 4. Materials and Methods

### 4.1. Insect Rearing

Thiamethoxam-sensitive melon aphids *A. gossypii* were transferred in September 2020 from Xueyan Shi’s lab at China Agricultural University, and were originally collected from a cucumber field in Nankou, Beijing, China (40°15′ N, 116°08′ E) in 2016. In the laboratory, cucumber seedlings were planted in 32-well plates with soil. Five melon aphids were gently transferred onto the leaves of cucumber seedlings with 2–3 real leaves for maintenance of the aphid strain. Aphids to be treated with dsRNA and TMX were put onto the cucumber leaves at the first instar and kept in a petri dish with a diameter of 60 mm until the end of the experiment. All melon aphids were reared in rooms with 75% relative humidity and a 16:8 h (light:dark) photoperiod, at 27 °C.

### 4.2. RNA Sequencing (RNA-Seq)

A total of 15 whole adult melon aphids were prepared for RNA sequencing analysis in a single sample. Three biological replications were prepared for aphids exposed to LC50 of TMX at 48 h and the water-treated control, respectively (refer to the methods in *Bioassay*). Total RNA was extracted with an RNAsimple Total RNA Kit (Tiangen Biotech, Beijing, China, DP419). The purity and concentration of extracted RNA was assessed using a NanoDrop One Microvolume UV-Vis Spectrophotometer (Thermo Fisher Scientific, Madison, WI, USA). An absorbance ratio of A260/280 of about 2.0 and A260/230 beyond 2.0 indicated a high quality of RNA. The integrity of RNA was evaluated by electrophoresis on 1% agarose gel. Qualified RNA with concentrations beyond 100 ng/μL was used for RNA sequencing. Transcript libraries were structured using the Illumina HiSeq 2000 sequencing platform(Illumina, San Diego, CA, USA). The genes with fold change >1.0 and FDR (false discovery rate) <0.05 were considered significantly differentially expressed.

### 4.3. Real-Time Quantitative PCR (qPCR)

Total RNA was isolated from each instar aphid, adult head, and adult trunk using an RNAsimple Total RNA Kit (Tiangen Biotech, DP419), following the manufactures. The quality and quantity of RNA was measured using an NanoDrop One Microvolume UV-Vis Spectrophotometer and agarose gel electrophoresis (Liuyi, Beijing, China, DYY-6C). Qualified RNA was used to synthesize first-stand cDNA using a FastQuant RT Kit (Tiangen Biotech, KR106-02). The template cDNAs were stored at −20 °C. Primers were designed based on the nucleotide sequence determined from NCBI (Appendix A). qPCR was performed using SYBR Green chemistry following the standard protocol of QuantStudio 6 Flex Real-Time PCR system (Thermo Fisher Scientific). The internal reference gene was *beta-actin*. The mean of three biological replicates and three technical replicates were used to estimate Ct values. The transcription level was calculated relative to the internal reference gene, following the 2^−∆∆CT^ method of Livak [75].

### 4.4. Phylogenetic Analysis and Sequence Alignment

The insect synapsin sequences were obtained from the National Center for Biotechnology Information (NCBI, https://www.ncbi.nlm.nih.gov, accessed on 30 October 2020), with the exception of the *Harmonia axyridis* TRINITY_DN46003_c0_g1 sequence, which was from the transcriptome analysis of *Harmonia axyridis* [76]. A phylogenetic tree of synapsin proteins was constructed using MEGA (version 7.0, Mega Limited, Auckland, New Zealand) with the neighbor-joining method and 1000 bootstrap replications.

### 4.5. dsRNA Synthesis

cDNAs synthesized from mRNA of the adult melon aphid using a 2×Taq PCR Master Mix kit (TIANGEN, KT201-02) were used to amplify a linear template for dsRNA synthesis (refer primer sequences of aphids in Appendix A). The amplified sequences were introduced into the pMD19T-Vector (Takara, Kyoto, Japan, 3271) and transformed into DH5α competent cells (Invitrogen, Carlsbad, CA, USA, BC102-02). The plasmids were extracted and verified by Sanger sequencing, then used as the template for dsRNA synthesis, using a T7 RiboMAX Express RNAi System (Promega, Madison, WI, USA, P1700). The T7 promoter sequence was incorporated with the aphid primer sequences for dsRNA synthesis (Appendix A) (Tsingke Biotechnology, Beijing, China). The EGFP-dsRNA, which targets the exogenous gene enhanced green fluorescent protein (EGFP), was used as a nonspecific control.

### 4.6. Bioassay

The first instar nymphs were gently put on cucumber leaves until the third instar for dsRNA assay. dsRNA was delivered onto the cuticle of these third instar aphids using the drop method (Figure 3B). To facilitate penetration through the insect cuticle, 0.1% volume of mild non-ionic surfactant alkyl polyglycoside (APG) (WanHua, Guangzhou, China) was added into the dsRNA solutions. The cationic nanocarrier SPc was mixed with dsRNA at a defined mass ratio of 1:1, as in previous studies [73,74]. A total amount of 0.1 μL solution including 250 ng (2500 ng/μL) or 500 ng syn*-*dsRNA (5000 ng/μL) with or without SPc was dropped onto the dorsal side of abdomen in third instar melon aphids. As negative controls, the same amount of EGFP-dsRNA was introduced onto tested insects and a solution containing the same amount of APG or SPc + APG was introduced onto the melon aphid. Total RNA was extracted 12 h and 24 h after the dsRNA delivery, for the analysis of endogenous *synapsin* mRNA level using qPCR. Twenty-four hours after dsRNA delivery, the behavior of the melon aphid was observed and recorded (Appendix A). As the melon aphid hardly moved, it was gently rolled over with a small soft brush until the abdomen was upward, and the course to right itself was recorded. The number of dead aphids was counted every 24 h, until 96 h after the dsRNA delivery, to evaluate the mortality rate. In addition, the number of offspring produced by the surviving melon aphids was recorded for three days in the adult stage. In the above biological assays, each group contained 10–15 aphids. Eight or more biological replicates were performed.

The toxicity of thiamethoxam (Simeiquan, Shenzhen, China, CAS: 153719-23-4) was evaluated using the leaf dip method, according to a previous study [77]. Thiamethoxam was dissolved in acetone and diluted with an equal volume of ddH_2_O, to concentrations of 64, 32, 16, 8, 4, 2, and 1 mg/L. The LC50 value at 48 h (5.24 mg/L) was calculated using probit analysis using GraphPad Prism 8.0.2 software (Appendix A). Solutions containing thiamethoxam and the comparison groups were sprayed onto the aphids (Figure 5A) in 4 replicates, and at least 20 adult aphids were used in each replicate. The volume of sprayed solution was 250 μL in each replicate. The mortality was assessed every day. The expression levels of verified genes were measured at 48 h, while *synapsin* was quantified every 12 h after the delivery of LC50 concentration. For the lower concentration of TMX (1 mg/L), *synapsin* expression was measured at 48 h. The final concentration of syn-dsRNA was 70 ng/μL in this spray assay.

### 4.7. Stability Test of Nanocarrier-Delivered dsRNA in Aphid Hemolymph

The methods to test the stability of nanocarrier-delivered dsRNA in aphid hemolymphs were described previously [72]. In detail, 50 adult aphids were punched and the hemolymph was extracted in 200 μL PBS and centrifuged at 2000× *g* rpm for 4 min at 4 °C. The supernatant was collected and centrifuged at 12,000× *g* rpm for 5 min at 4 °C, to remove the tissue impurities. The supernatant after the second centrifugation was 100% hemolymph. In the degradation assays, different dilutions of hemolymph were prepared and mixed with syn-dsRNA. The mixture was incubated at 25 °C for 3 h and then analyzed by electrophoresis on a 1% agarose gel.

### 4.8. Statistical Analysis

For the relative expression of *synapsin* in each instar of the melon aphid, the gene expression in the first instar was normalized to 1. The expression in adult heads was normalized to that in adult trunks. After the dsRNA delivery, the expression in the syn-dsRNA-treated insects was normalized to the EGFP-dsRNA-treated control. In exposure to TMX, the expression was normalized to the corresponding control. The mortality was the quotient of the total number of dead insects divided by the initial total number of tested insects. The data were analyzed, and statistical charts were produced using GraphPad Prism (version 8.0, San Diego, CA, USA). Nonparametric Student’s *t*-test was used for pairwise comparison (* *p* < 0.05, ** *p* < 0.01, and *** *p* < 0.001), and nonparametric ANOVA followed by the Tukey’s multiple comparison test was used for multiple comparisons (*p* < 0.05). The error bars indicate SEM.

## 5. Conclusions

In summary, we identified a new essential gene, *synapsin,* in melon aphids. Knockdown of the expression of *synapsin* by dsRNA was lethal to melon aphids. The co-administration of syn-dsRNA and thiamethoxam effectively resulted in the death of melon aphids, and the mortality could be greatly improved in combination with the nanocarrier star polycation. These results provide new ideas for the selection of an RNAi strategy for green pest management of the melon aphid.

## Figures and Tables

**Figure 1 ijms-23-09388-f001:**
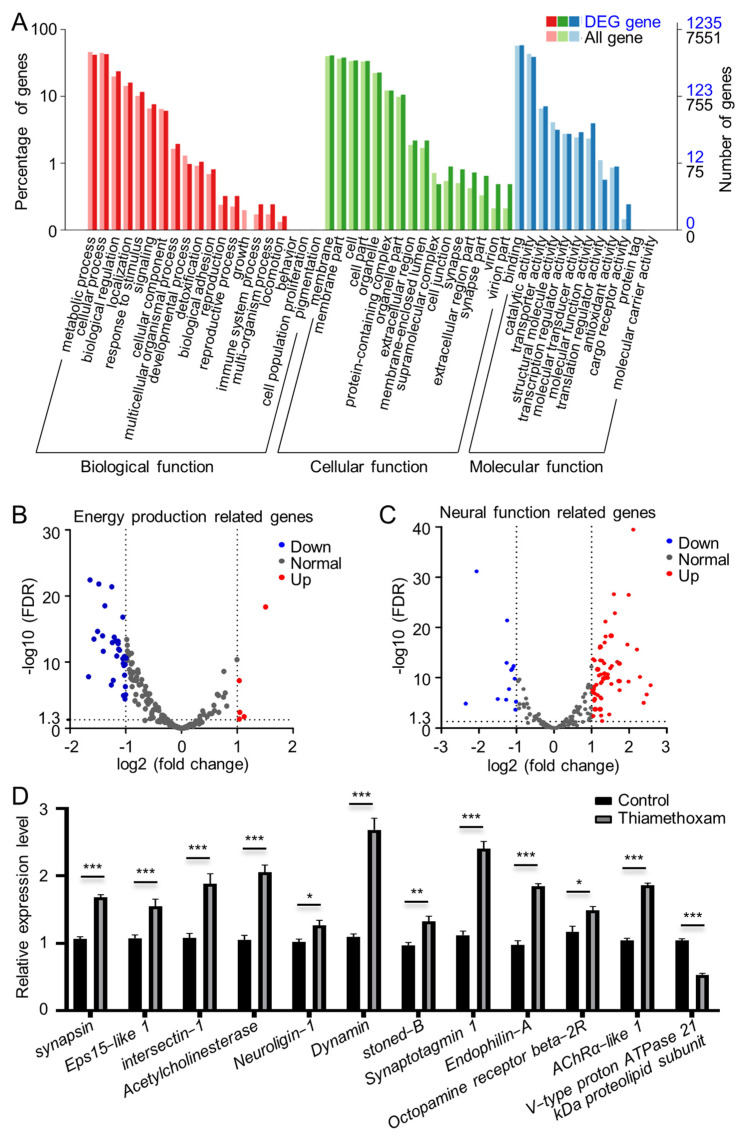
Genes were differentially expressed by exposure to TMX. (**A**) Analysis of DEGs with GO annotation. (**B**) Volcano plots of genes in energy production. (**C**) Volcano plots of genes in neural function. (**D**) qPCR confirmed the changes of the neurofunctional genes. * indicates significance with pairwise comparison of gene expression in the control and TMX treated aphids (Student’s *t*-test, *** *p* < 0.001, ** *p* < 0.01, and * *p* < 0.05).

**Figure 2 ijms-23-09388-f002:**
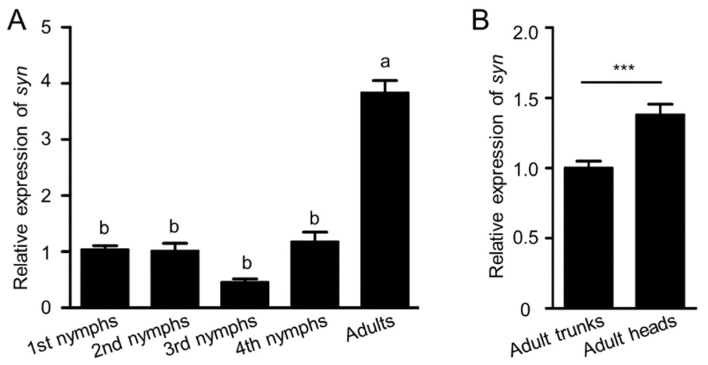
Assessment of the relative gene expression of *synapsin* in various development stages of the melon aphid. (**A**) Expression profiles of *synapsin* at different stages. The relative expression of *synapsin* in the first instar nymph was normalized to 1. The letters above each column indicate significant differences (Tukey’s multiple comparison test, *p* < 0.05). (**B**) The expression profile of *synapsin* in different tissues. *** indicates the significance by pairwise comparison (Student’s *t*-test, *** *p* < 0.001).

**Figure 3 ijms-23-09388-f003:**
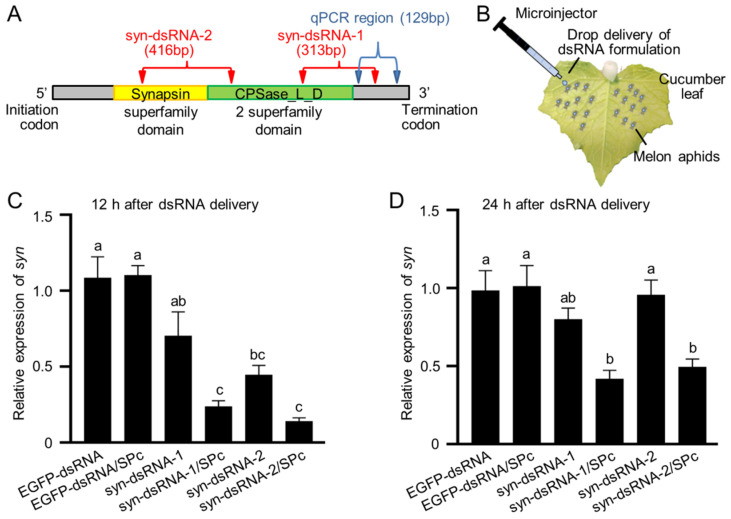
*synapsin* expression was regulated by syn-dsRNA. (**A**) The predicted protein structure of *synapsin* in the melon aphid. The syn-dsRNA targeting sites and qPCR amplicon are indicated. (**B**) Schematic diagram of transdermal delivery system of dsRNA formulations. (**C**) *synapsin* was downregulated by syn*-*dsRNA at 12 h after dsRNA delivery. (**D**) *synapsin* was downregulated in the presence of SPc at 24 h after dsRNA delivery. The letters above each column indicate significant differences (Tukey’s multiple comparison test, *p* < 0.05).

**Figure 4 ijms-23-09388-f004:**
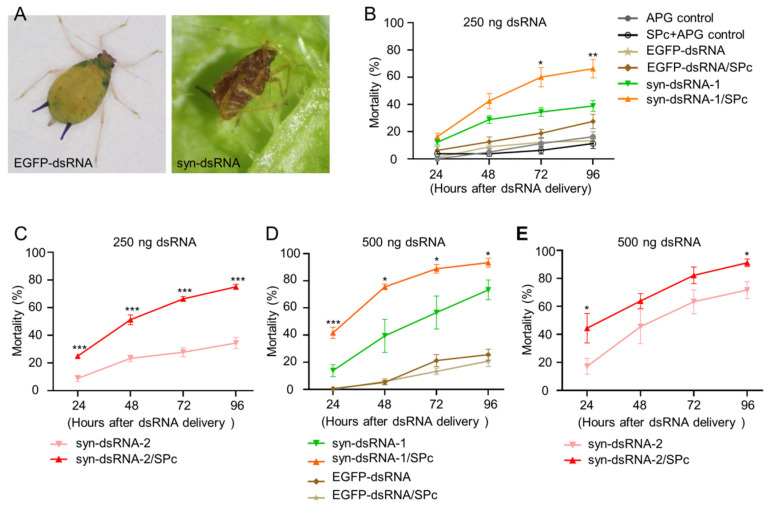
The lethal effect of syn*-*dsRNA. (**A**) An example of a viable melon aphid treated with EGFP-dsRNA and one treated with syn-dsRNA. (**B**,**C**) The delivery of 250 ng syn-dsRNA resulted in lethality to melon aphids. (**D**,**E**) The delivery of 500 ng syn-dsRNA resulted in lethality. SPc enhanced the mortality caused by syn-dsRNA. * indicates the significance by pairwise comparison of mortality between syn dsRNA/SPc and non-SPc treated aphids at each time point (Student’s *t*-test, *** *p* < 0.001, ** *p* < 0.01, and * *p* < 0.05).

**Figure 5 ijms-23-09388-f005:**
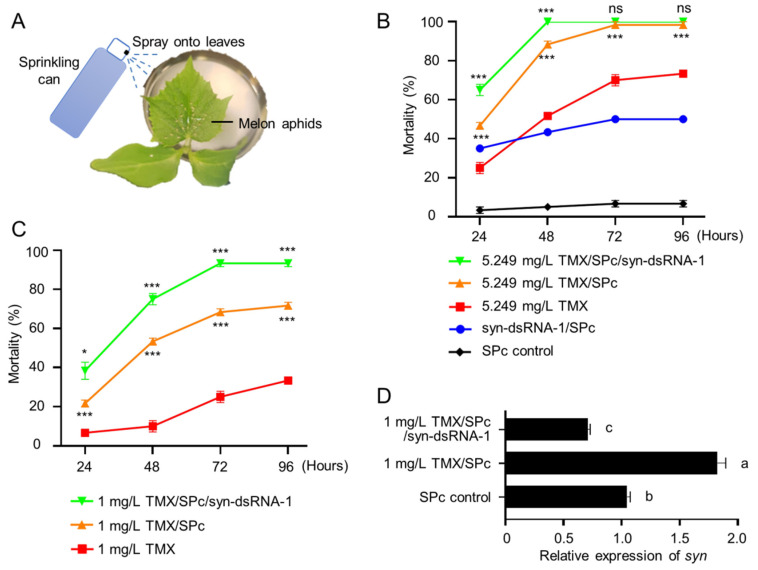
Co-delivery of syn-dsRNA and TMX enhanced the lethality rate. (**A**) Schematic diagram of spraying the insecticide. (**B**) syn-dsRNA and TMX had a synergistic effect at early time points. (**C**) syn-dsRNA and TMX had a synergistic effect at low levels of TMX. In (**B**,**C**), * above the green dot indicates significance by pairwise comparison of the mortality between TMX/SPc/syn*-*dsRNA and non-SPc treated aphids at each time point, while * underneath the orange dot is a comparison between TMX/SPc and TMX samples (Student’s *t*-test, *** *p* < 0.001 and * *p* < 0.05). ns, no significance. (**D**) *synapsin* was decreased in combination of syn-dsRNA and TMX. The letters on the right side indicate significant differences (Tukey’s multiple comparison test, *p* < 0.05).

**Figure 6 ijms-23-09388-f006:**
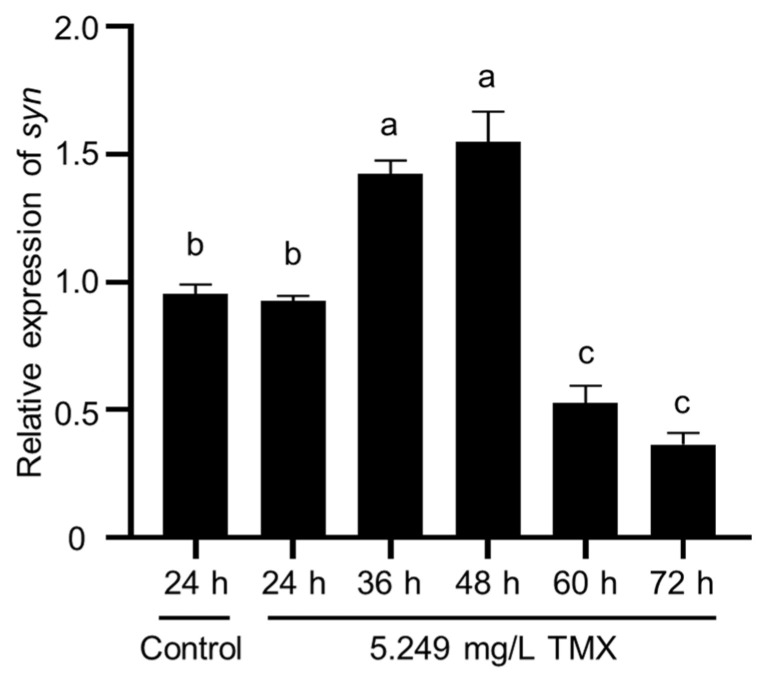
The expression levels of *synapsin* were changed during exposure to TMX. The letters above each column indicate significant differences (Tukey’s multiple comparison test, *p* < 0.05).

## Data Availability

All data are contained within the article.

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
