# Peer review of "The Synergistic Effect of Thiamethoxam and Synapsin dsRNA Targets Neurotransmission to Induce Mortality in Aphis gossypii"

_ijms, 2022, doi:10.3390/ijms23169388_

Round 1

Reviewer 1 Report

The manuscript provides sufficient information on the scientific background, results, tabular and graphic expression, imaging and video. Also, the references are comprehensively and correctly associated and uniformly described.

However, I would have some recommendations related to the methodology and conclusions, as follows:

To Materials and Methods

Before lines 286-289: I suggest a description that suggests a placement in time and space of the research. The manuscript does not show the location (which laboratory, institution, university?) but neither the period of studies.

 Lines 286-289: It would be a good idea to extend the ideas related to Insect rearing to bring to your attention the amount of samples (number of aphids, number of cucumber leaves) and container documents or samples you used. Provide details about this stage of organization so that the reader has a clear opinion about the methodology.

The conclusions are missing, so I recommend you to extract 3-4 brief ideas about the very good, very detailed and quality results in my judgment. Why? It would be useful for the reader and would clarify the essence of the numerous data  and information.

Reviewer 2 Report

Dear Authors,

Unfortunatelly I have to reject your manuscript. I have big doubts about the quality of the experimental data you have received. Besides, the article requires additional experiments. Below you will find my comments on the main problems for your manuscript.

1. Describe what you mean by gene and unigen in section 2.1

2. For qPCR you used just one reference gene, it's too little! Provide 3 reference gene at least.

3. Nanodrop is not suitable for measuring RNA quality, you can estimate this parameter using agarose gel (very rough estimate) or RNA analyser. RIN (RNA integrity number) is Đ° generally accepted assessment of RNA quality. For RNA sequencing it should be >= 7.

4. There are 2 figures of gels in the Supplementary Materials. I did not find references to these figures (S2A and S2B) in the text of the manuscript, , moreover, it is absolutely not clear what is on these gels and what it refers to.

Round 2

Reviewer 2 Report

Dear Authors,

Thanks for you corrections in the manuscript. Now I thonk it could be accepted.